# Living Donor Liver Re-Transplantation for Recurrent Hepatoblastoma in the Liver Graft following Complete Eradication of Peritoneal Metastases under Indocyanine Green Fluorescence Imaging

**DOI:** 10.3390/cancers11050730

**Published:** 2019-05-26

**Authors:** Nobuhiro Takahashi, Yohei Yamada, Ken Hoshino, Miho Kawaida, Teizaburo Mori, Kiyotomo Abe, Takumi Fujimura, Kentaro Matsubara, Taizo Hibi, Masahiro Shinoda, Hideaki Obara, Kyohei Isshiki, Haruko Shima, Hiroyuki Shimada, Kaori Kameyama, Yasushi Fuchimoto, Yuko Kitagawa, Tatsuo Kuroda

**Affiliations:** 1Department of Pediatric Surgery, Keio University School of Medicine, Tokyo 160-8582, Japan; tkhsnbhr430@gmail.com (N.T.); hoshino@z7.keio.jp (K.H.); 0105mori@keio.jp (T.M.); matuzono300@yahoo.co.jp (K.A.); fujitaku3@gmail.com (T.F.); yfuchimoto@keio.jp (Y.F.); kuroda-t@z8.keio.jp (T.K.); 2Department of Pediatric Surgery, National Center for Child Health and Development, Tokyo 157-0074, Japan; 3Department of Pathology, Keio University School of Medicine, Tokyo 160-8582, Japan; m.kawaida@a2.keio.jp (M.K.); kameyama@a5.keio.jp (K.K.); 4Department of Surgery, Keio University School of Medicine, Tokyo 160-8582, Japan; kmatsubaravs@gmail.com (K.M.); taizohibi@gmail.com (T.H.); masa02114@yahoo.co.jp (M.S.); obara@z3.keio.jp (H.O.); kitagawa@a3.keio.jp (Y.K.); 5Department of Pediatric Surgery and Transplantation, Kumamoto University Graduate School of Medical Sciences, Kumamoto 860-0862, Japan; 6Department of Pediatrics, Keio University School of Medicine, Tokyo 160-8582, Japan; scratchbeetle@hotmail.com (K.I.); sharuko@keio.jp (H.S.); hshimada@a5.keio.jp (H.S.); 7Children’s Cancer Center, National Center for Child Health and Development, Tokyo 157-0074, Japan; 8Department of Pediatric Surgery, International University of Health and Welfare, Chiba 286-0048, Japan

**Keywords:** hepatoblastoma, indocyanine green, living donor liver transplantation, transplant oncology

## Abstract

The curability of chemotherapy-resistant hepatoblastoma (HB) largely depends on the achievement of radical surgical resection. Navigation techniques utilizing indocyanine green (ICG) are a powerful tool for detecting small metastatic lesions. We herein report a patient who underwent a second living donor liver transplantation (LDLTx) for multiple recurrent HBs in the liver graft following metastasectomy for peritoneal dissemination with ICG navigation. The patient initially presented with ruptured HB at 6 years of age and underwent 3 liver resections followed by the first LDLTx with multiple sessions of chemotherapy at 11 years of age. His alpha-fetoprotein (AFP) level increased above the normal limit, and metastases were noted in the transplanted liver and peritoneum four years after the first LDLTx. The patient underwent metastasectomy of the peritoneally disseminated HBs with ICG navigation followed by the second LDLTx for multiple metastases in the transplanted liver. The patient has been recurrence-free with a normal AFP for 30 months since the second LDLTx. To our knowledge, this report is the first successful case of re-LDLTx for recurrent HBs. Re-LDLTx for recurrent HB can be performed in highly select patients, and ICG navigation is a powerful surgical tool for achieving tumor clearance.

## 1. Introduction

The introduction of cisplatin-based chemotherapy has provided significant survival benefits for patients with metastatic hepatoblastoma (HB) [1]; however, chemotherapy-resistant tumors can only be cured by complete surgical resection. Surgical management of metastases is known to improve a patient’s prognosis regardless of their sensitivity for chemotherapy, with a goal of clearing away all metastatic lesions [2]. Indocyanine green (ICG) is a reagent widely used to evaluate the liver function before hepatectomy. Recently, several series of navigation surgery using ICG for detecting small metastatic HBs were reported [3,4,5,6]. This technique involves the selective uptake of ICG by hepatocytes and HB cells, which allows for the selective visualization of such lesions in near-infrared mode.

We herein report the first application of ICG navigation surgery for peritoneal dissemination of HBs for a patient who suffered from chemotherapy-resistant recurrence of HBs four years after the first rescue living donor liver transplantation (LDLTx). Simultaneously, the patient was found to have multiple metastases in the transplanted liver, presumably derived from peritoneally disseminated tumors. We conducted sequential transplantation involving clearing of the dissemination by ICG navigation surgery followed by re-transplantation from a living donor. To our knowledge, this report describes the first case of successful re-transplantation for recurrent HBs in the transplanted liver. The potential application of ICG navigation surgery along with LTx and the indications for re-transplantation for recurrent HBs are discussed.

## 2. Case Report

The patient presented with hypovolemic shock due to the rupture of HB when he was 6 years old. The initial stage was Pretreatment Tumor Extent (PRETEXT) II (V0, P0, E0, F0, R1, C0, N0, M0) [7]. He underwent right hepatic artery embolization and chemotherapy consisting of cisplatin (80 mg/m^2^) and tetrahydropyranyl adriamycin (THP-ADR) (30 mg/m^2^) followed by right lobectomy based on the protocol described in Japanese Study Group for Pediatric Liver Tumor (JPLT)-1 [8]. The initial histological analysis revealed HB without features of hepatocellular carcinoma. Two adjuvant cycles of the same regimen were added postoperatively. However, as the HB recurred in the remnant of the liver a year later (at 7 years of age), the patient underwent partial resection followed by an additional 4 cycles of the same regimen.

Unfortunately, the tumor recurred in the remaining lobe of the liver, so partial resection was performed again when he was 8 years old. Postoperatively, 4 cycles of the C5V regimen (cisplatin (90 mg/m^2^), 5-fluorouracil (600 mg/m^2^) and vincristine (1.5 mg/m^2^)) were provided. At 9 years of age, magnetic resonance imaging (MRI) revealed the recurrence of HB in the liver, so the patient was referred to our center and underwent living donor liver transplantation (LDLTx) as a rescue treatment. Irinotecan (CPT-11) was selected as an adjuvant therapy after LDLTx. The details of his treatment course and AFP values are shown in Figure 1. A histological analysis revealed wholly epithelial-type (fetal subtype) HB, intrahepatic metastasis(im)(+), s0, vp1, vv0, va0, b0 and sm(−). The postoperative course was uneventful, and the patient was discharged after a month.

His AFP remained within normal range for 45 months after the first LDLTx and then began to rise without any precedent events when he was 14 years old. MRI showed nodules in the transplanted liver, which prompted us to perform exploratory laparotomy. The pathological findings of metastatic lesion in the liver were HB, which consistent with the original histology at the time of the first LDLTx (Figure 2A,B). Elastica van Gieson (EVG) stain of the specimen is also shown, indicating the HB surrounded by elastic fiber of portal veins (Figure 2C). Intraoperatively, we found peritoneal nodules other than metastatic lesions in the liver that turned out to be metastatic HBs (Figure 2D,E).

We then planned to perform ICG navigation surgery to achieve complete eradication of the disseminated HBs several days later. The study was approved by the institutional ethical review board of Keio University School of Medicine (approval No. 20160226). ICG (0.5 mg/kg) was given 72 h prior to the operation to minimize the background uptake by the normal hepatocytes. A Photodynamic Eye system^®^ (PDE^®^; Hamamatsu Photonics, Hamamatsu, Japan) was used to visualize the lesions taking up ICG in near-infrared mode. While the exact locations of the disseminated HBs were not clearly identified in white-light mode (Figure 3A,C), the corresponding view in near-infrared mode (Figure 3B,D) showed well-demarcated nodules in the parietal peritoneum and mesocolon adjacent to the transplanted liver, all of which were successfully excised.

All specimens that were visualized by PDE^®^ were histologically positive for HB tissues. Thereafter, meticulous examination of the entire abdominal cavity by PDE^®^ was performed, which revealed no other metastatic lesions besides the aforementioned area in the right upper quadrant. However, the transplanted liver graft was infiltrated with diffuse metastatic HBs. After a two-month interval, another laparotomy utilizing ICG navigation (0.5 mg/kg of ICG was administered three days before laparotomy) was performed to explore whether or not there were any residual disseminated HBs. Although numerous uptakes of ICG in the transplanted liver were noted, we confirmed the complete absence of any extrahepatic lesions in the abdomen.

In a multidisciplinary conference, we discussed whether or not, provided the all detectable extrahepatic metastases were brought under control, re-transplantation is rational from an oncological perspective and is the only curative option. In Japan, since the number of deceased donor transplantations is limited (50–60 liver transplantations from deceased donors per year), a second LDLTx was considered. The patient and the family were well informed of the risk of recurrence even after a second LDLTx but remained willing to proceed. The institutional review board approved the second LDLTx.

Prior to the second LDLTx, CPT-11 (20 mg/m^2^) was administered to delay the growth of the tumor cells. The AFP value decreased from 439 ng/ml to 297 ng/ml after 2 cycles of CPT-11. The second LDLTx was performed when the patient was 14 years old, which was 62 months after the first LDLTx. Again, 0.5 mg/kg of ICG was administered intravenously three days prior to the second LDLTx. Fortunately, no extrahepatic tumor was observed at the time of the second LDLTx, although multiple instances of the uptake of ICG in the explanted liver were visualized by PDE^®^ (Figure 4A,B). Of note, the hepatectomy of the donor was delayed until it was confirmed that no new extahepatic metastases were seen by PDE^®^. The ICG-positive lesions in the liver were compatible with HBs pathologically. Postoperatively, an additional two cycles of CPT-11 were provided. The patient has been recurrence-free for 30 months since the second LDLTx, with the value of AFP being within the normal limit (Figure 1).

## 3. Discussion

A surgical approach has shown marked efficacy for the treatment of recurrent HBs, which are considered highly chemo-resistant [9]. A growing number of successful metastasectomies for recurrent pulmonary metastases of HBs and a subsequent tumor-free survival have been reported [10,11,12]; however, little is known about the survival benefit in the eradication of peritoneal dissemination or the role of re-transplantation for recurrence in a transplanted liver graft.

When considering the indication of re-transplantation for this patient, three factors were discussed thoroughly in the multidisciplinary team meeting. First, the fact that this patient initially presented with ruptured HBs at six years of age led to the hypothesis that the disseminated HBs did not develop after the first LDLTx but existed from the beginning. They had not been eradicated through multiple operations and chemotherapy and subsequently became chemo-resistant. This speculation was compatible with the operative finding that the dissemination was localized in the upper-right of the abdominal cavity, as confirmed through two exploratory laparotomies and the second LDLTx procedure. As HB cases that develop accompanying tumor rupture tend to recur more frequently and result in a poorer prognosis than those without tumor rupture [7], the patient should have been treated with a more intensive regimen according to the current risk stratification [13]. Alternatively, intraperitoneal chemotherapy, which has a role in the treatment of peritoneal metastasis of hepatocellular carcinoma, might be indicated in cases in which intravenous chemotherapy is ineffective [14]. If ICG navigation was available at the time of the first resection, it might have helped surgeons identify the presence, location and extent of peritoneal disease, and the lesion with the uptake of ICG should have been sent to pathology.

Second, the absence of lung metastases and the timing of recurrence after the first LDLTx procedure were interesting. The tumor recurred after an interval of four years following the first LDLTx procedure and rapidly spread as multiple hepatic metastases. To our knowledge, the longest interval between LTx and the recurrence of HB was 2.8 years, as reported in a Japanese national survey [15]. These findings indicate that the recurrent tumors grew very slowly and were localized in the abdominal cavity, which favors our proposal for re-transplantation. If the tumor recurs in the graft or lungs soon after the first LTx, which indicates that the tumor is growing quickly and thus is now spreading throughout the entire body, then such a patient would not be indicated for additional surgery. Although it remains a matter of debate how long an interval is necessary before a patient may be considered indicated for re-transplantation for recurrent HBs, we believe that at least a two-year period of normal AFP levels after the first LTx should be required. Regarding the time between the clearing of metastatic disease with ICG guided surgery and the second transplantation, we set a period of two months arbitrarily between the first and second exploratory laparotomy procedures. Then, the second LDLTx was scheduled one month later. Although re-transplantation must be performed before multiple liver metastases spread further, we considered it prudent to confirm the absence of newly developed peritoneal disease for several months before moving forward. In addition, although we did not check for the presence of circulating tumor cells in this specific patient, the presence of circulating tumor cells seems to be associated with a poor prognosis in patients with hepatocellular carcinoma [16]. With recent technical advances, the absence of circulating tumor cells in HB patients may become an objective tests to justify liver transplantation.

Third and most importantly, the technical feasibility of eradicating the HBs was discussed. In this regard, ICG navigation surgery played an essential role. ICG, which is widely used to assess the liver function, is selectively taken up by HBs as well as normal hepatocytes but excreted in a delayed fashion from HB tissue. Due to this delayed excretion, the specific visualization of both primary and metastatic HBs in the surgical field is possible. However, several limitations associated with this technique should be mentioned. The false-positive rate with ICG navigation surgery is reported to be 10–20% (number of lesions that were not pathologically diagnosed as HBs among ICG-positive lesions). Furthermore, in surveys of hepatocellular carcinoma, the sensitivity and specificity were reported to vary depending on the histology, liver function and timing of surgery after ICG injection [17,18,19,20]. In addition, the fluorescence produced by ICG can only be detected when the lesions are located 5–10 mm from the surface. Further studies will be needed in order to address these issues. Despite these limitations, since ICG navigation enables the detection of nodules <0.1 mm in diameter and facilitates the clear demarcation of tumors [3,5], we believe that this technology is the best available to present to detect HBs. Our patient is living proof of the emerging paradigm of transplant oncology, the fusion of transplantation medicine and oncology to open new horizons in cancer treatment [21].

Based on the findings and the discussions mentioned above, our hypothesized mechanism of the recurrence that developed in this patient is presented in Figure 5. We propose the concept of “sequential transplantation”, which is the complete elimination of peritoneal dissemination using ICG navigation followed by the replacement of the affected liver. The fact that ICG navigation allowed us to remove the nodules that were otherwise invisible to the naked eye despite being pathologically proven disseminated HBs and that the patient has been tumor-free for 2.5 years since the second LDLTx provides proof of this principle. As it might be too early to conclude that the patient is cured, regular surveillance is mandatory and switching the immunosuppression regimen from tacrolimus to everolimus is now being considered in the anticipation of a potential antitumor effect.

## 4. Conclusions

In conclusion, re-LDLTx for multiple intra-graft HB metastases can be indicated for selected patients. ICG navigation provides the best surgical tool currently available for achieving complete eradication of HBs. Yet, as this case is the first attempted re-transplantation for recurrent HB, a prudent approach should be taken when considering the concept of sequential transplantation given the precious donor source.

## Figures and Tables

**Figure 1 cancers-11-00730-f001:**
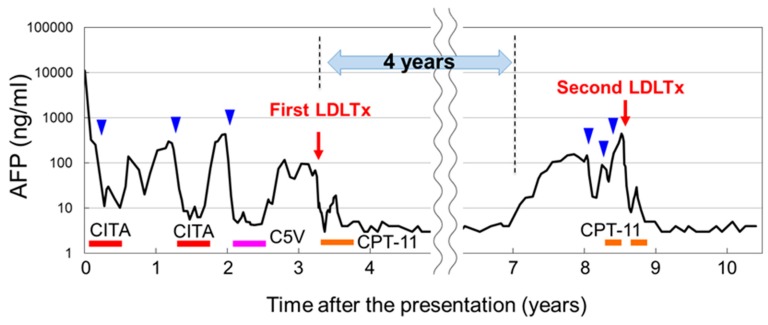
The clinical course of the patient. The years after the initial presentation are plotted on the horizontal axis, and the values of alpha-fetoprotein (AFP) are plotted on the vertical axis. Blue arrowheads indicate laparotomies other than living donor liver transplantation (LDLTx). Bars on the bottom represent the chemotherapeutic regimen. CITA: cisplatin 80 mg/m^2^ and tetrahydropyranyl adriamycin (THP-ADR) 30 mg/m^2^, C5V: cisplatin 90 mg/m^2^, 5-fluorouracil 600 mg/m^2^ and vincristine 1.5 mg/m^2^, CPT-11: Irinotecan 20 mg/m^2^.

**Figure 2 cancers-11-00730-f002:**
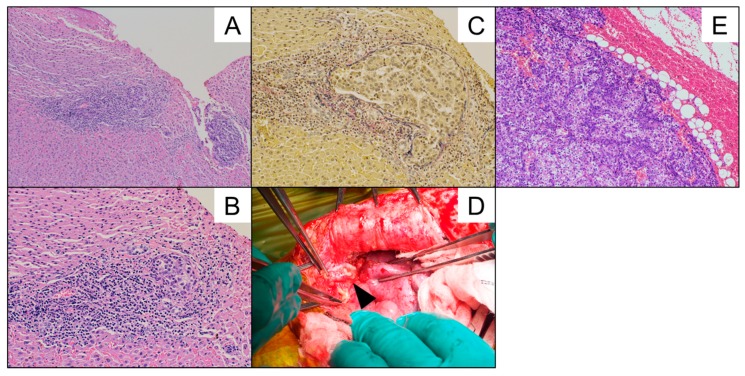
(**A**,**B**,**C**) The pathological findings of the metastatic nodule in the liver, which were compatible with wholly epithelial-type (fetal subtype) hepatoblastoma (HB) with vascular invasion. (**A**,**B**; H.E. stain, **A**; 100×, **B**; 200×, **C**; Elastica van Gieson (EVG) stain, 200×). (**D**) The arrowhead represents the peritoneal nodule adjacent to the transplanted liver, which was noted in the first laparotomy after the first LDLTx procedure. (**E**) The pathological findings of the peritoneal nodule is shown in Figure (H.E. stain, 100×).

**Figure 3 cancers-11-00730-f003:**
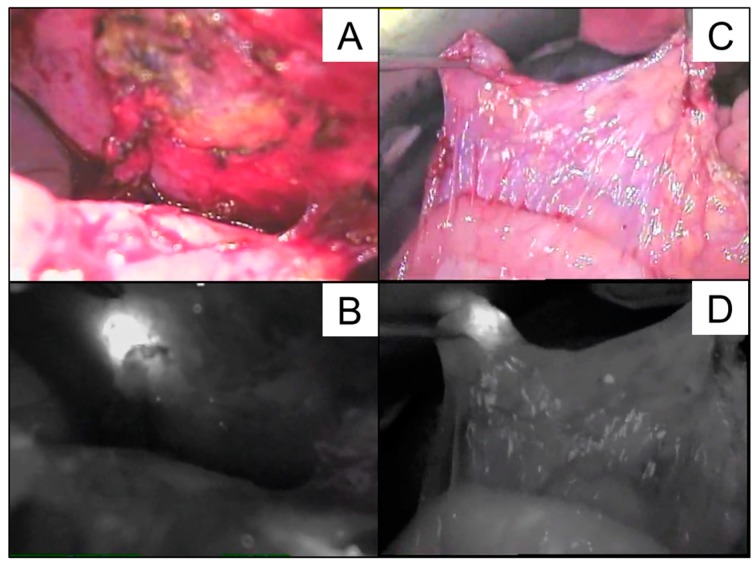
White-light mode (**A**,**C**) and corresponding near-infrared mode (**B**,**D**) findings in the peritoneal cavity at the time of the second laparotomy after the first LDLTx procedure are shown, macroscopic image.

**Figure 4 cancers-11-00730-f004:**
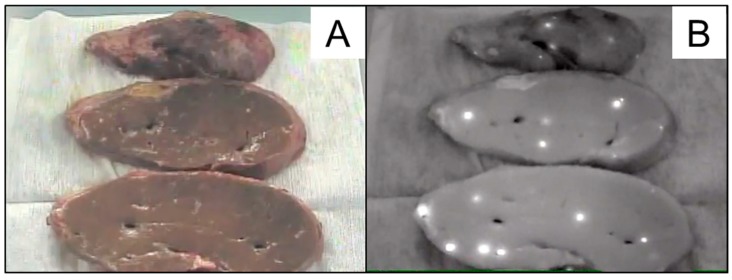
(**A**) The sliced explanted liver in white-light mode and (**B**) near-infrared mode. The hot spots in near-infrared mode were compatible with hepatoblastomas in histology, macroscopic image.

**Figure 5 cancers-11-00730-f005:**
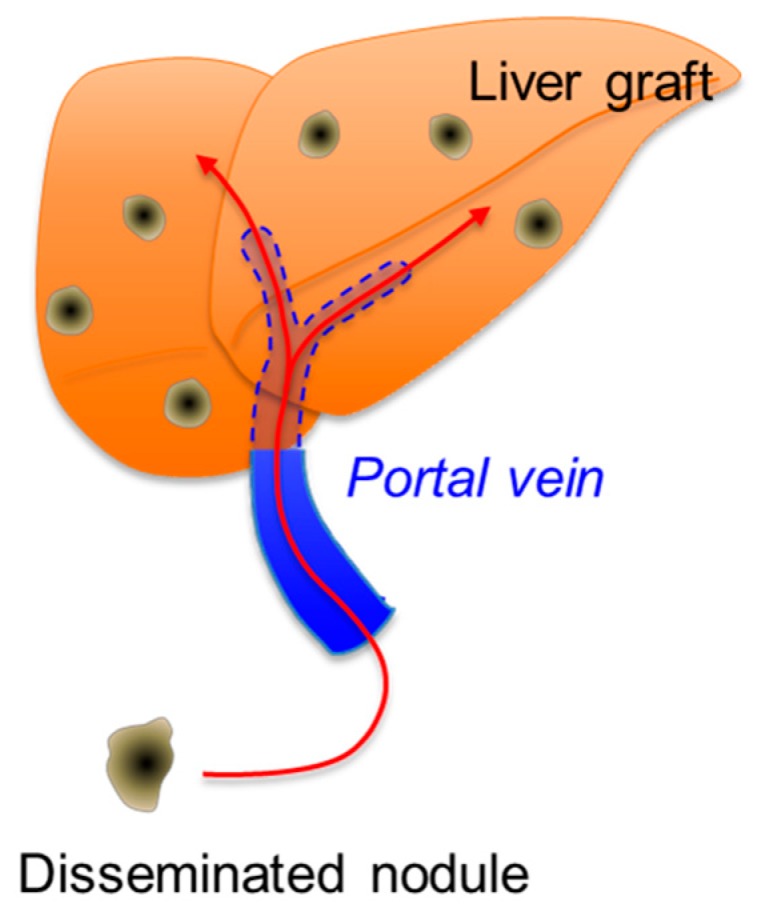
A schematic illustration of the hypothesized mechanism of recurrence in this patient. Multiple metastases in the liver graft are assumed to be derived from disseminated nodules through the portal vein.

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
