# Peer review of "Living Donor Liver Re-Transplantation for Recurrent Hepatoblastoma in the Liver Graft following Complete Eradication of Peritoneal Metastases under Indocyanine Green Fluorescence Imaging"

_cancers, 2019, doi:10.3390/cancers11050730_

Round 1

Reviewer 1 Report

This report describes the use of ICG Navigation to map occult sites of peritoneal metastasis in order for surgical removal and second transplantation. The use of ICG in this context is novel and of interest to surgeons and oncologists who care for this rare disease. The methodology used and conclusions are sound.

The age of onset of this patient raises the possibility that this patient might initially have harbored a hepatocellular neoplasm NOS / hepatoblastoma w/ HCC features. Further elucidation of the original histology would be useful. As the authors point out, the initial rupture at diagnosis suggests that, in the current CHIC classification system, the patient would have been higher risk and have received more intensive adjuvant chemotherapy initially. This might have obviated the need for subsequent surgeries although that is speculative.

The concept of Staged transplantation, while interesting, should be described as having utility in only such specialized situations as this one. Different national transplantation organizations have different criteria / levels iof permissiveness for use of OLT/ retranplantation and this should also be stated more clearly in the discussion.

Author Response

This report describes the use of ICG Navigation to map occult sites of peritoneal metastasis in order for surgical removal and second transplantation. The use of ICG in this context is novel and of interest to surgeons and oncologists who care for this rare disease. The methodology used and conclusions are sound.

The age of onset of this patient raises the possibility that this patient might initially have harbored a hepatocellular neoplasm NOS / hepatoblastoma w/ HCC features. Further elucidation of the original histology would be useful. As the authors point out, the initial rupture at diagnosis suggests that, in the current CHIC classification system, the patient would have been higher risk and have received more intensive adjuvant chemotherapy initially. This might have obviated the need for subsequent surgeries although that is speculative.

The concept of Staged transplantation, while interesting, should be described as having utility in only such specialized situations as this one. Different national transplantation organizations have different criteria / levels iof permissiveness for use of OLT/ retranplantation and this should also be stated more clearly in the discussion.

We deeply appreciate your comments.

The initial histological description (no features of HCC) was added in line 63, however it was obtained post-chemotherapy because the patient presented with hypovolemic shock due to ruptured HB in a local hospital.

Regarding the indication of staged transplantation, the limitation of the use of deceased donors is mentioned in line 101 and additional sentences have been added in lines 205-207.

Reviewer 2 Report

This is an interesting case report regarding the management of hepatoblastoma that had ruptured originally by resection, followed by LDLT and then 4yrs later when recurrence was detected with a combination of targeted resection of peritoneal metastatic lesions and LDLT. The merit of this paper lies in the use of the targeted resection with ICG as a very interesting application, as well as in the discussion of the biological behavior of the tumor (recurrence seen only 4 yrs later). Could the authors please respond to the following comments:

a) Having used LDLT twice for a patient with an aggressive tumor has to be approached very carefully given the fact that we are placing two living donors (which is a very precious resource) in danger. The authors may wish to discuss the possibility of an extended criteria cadaveric donor, although obviously in different countries there are cultural issues that may make this difficult

b) Could the authors discuss the issue of immunosuppression used and potential use of antitumor immunosuppressant agents such as rapamycin?

c) Is there a proposed time period between the clearing of the metastatic disease with ICG guided surgery and the transplantation?

Author Response

This is an interesting case report regarding the management of hepatoblastoma that had ruptured originally by resection, followed by LDLT and then 4yrs later when recurrence was detected with a combination of targeted resection of peritoneal metastatic lesions and LDLT. The merit of this paper lies in the use of the targeted resection with ICG as a very interesting application, as well as in the discussion of the biological behavior of the tumor (recurrence seen only 4 yrs later). Could the authors please respond to the following comments:

a) Having used LDLT twice for a patient with an aggressive tumor has to be approached very carefully given the fact that we are placing two living donors (which is a very precious resource) in danger. The authors may wish to discuss the possibility of an extended criteria cadaveric donor, although obviously in different countries there are cultural issues that may make this difficult

In Japan, the number of deceased donor transplantations is limited (50-60 liver transplantations per year). Thus, given the risk of recurrence after the transplantation deceased donor transplantation was not indicated in this case. (Lines 101-103 were added)

b) Could the authors discuss the issue of immunosuppression used and potential use of antitumor immunosuppressant agents such as rapamycin?

Some centers favor everolimus (covered by national health insurance in Japan since 2018) for liver transplantation for adult patients with hepatocellular carcinoma in anticipation of anti-tumor effects. We have commented on this issue in the discussion. (lines 194-197 were added)

c) Is there a proposed time period between the clearing of the metastatic disease with ICG guided surgery and the transplantation?

I would like to thank the reviewer for raising this issue. We waited an arbitrary period of 3 months between the clearing of metastatic disease with ICG-guided surgery and transplantation. The second transplant was performed 14 months after the AFP level began to rise. Although we have to admit that there was no evidence to support that such an interval was long enough to prove the successful eradication of the metastatic disease, we made every effort to confirm the absence of metastatic diseases repetitive exploration with ICG navigation at certain intervals and to justify re-transplantation. (lines 162-168)

Reviewer 3 Report

I enjoyed your case report and it certainly gives some hope to treating long term even locally metastatic and recurrent Hepatoblastoma with liver transplantation. I am though reluctant to call the patient cured as it took 4 years for the recurrent disease to clearly manifest and it has only been 2.5 years since his transplant. I also want to point out that the cancer does not seem to be resistant to CPT-11 as the AFP rose after both transplants and fell to normal levels after a short course of that chemotherapy (reading into Figure 1).

I have a few questions:

Why do you feel optimistic now about the possibility of having cured your patient?

In retrospect on the index patient, were the liver recurrences requiring a second and third resection due to new metastatic lesions from the presumed peritoneal disease, or was it due to inadequate resection, in effect?

In your next patient who presents with ruptured hepatoblastoma, in light of finding peritoneal spread of the cancer 4 years after the first liver transplant, would you consider using ICG to find and remove/destroy peritoneal implants at the time of the liver resection or probably better at the time of reresection or, in view of the likelihood of having tiny metastatic implants too small to see, would you consider doing intraperitoneal chemotherapy at the time of liver resection?

One purported cause of recurrent cancer is that it may be in the blood and be hanging out in the bone marrow. Did you ever check for HB cells in your patient's blood?

I would like those questions addressed in your paper.

Fascinating case.

Author Response

I enjoyed your case report and it certainly gives some hope to treating long term even locally metastatic and recurrent Hepatoblastoma with liver transplantation. I am though reluctant to call the patient cured as it took 4 years for the recurrent disease to clearly manifest and it has only been 2.5 years since his transplant. I also want to point out that the cancer does not seem to be resistant to CPT-11 as the AFP rose after both transplants and fell to normal levels after a short course of that chemotherapy (reading into Figure 1).

I have a few questions:

Why do you feel optimistic now about the possibility of having cured your patient?

After the each metastasectomy (indicated by blue triangles), the AFP values declined but soon began to rise, which suggests the presence of residual hepatoblastoma. It has been 2.5 years (although there is no scientific evidence regarding the length of time required to declare that the patient is “cured”) since the transplant when the patient had a normal AFP value. We hope that the HB was eradicated. (lines 194-197 were added)

In retrospect on the index patient, were the liver recurrences requiring a second and third resection due to new metastatic lesions from the presumed peritoneal disease, or was it due to inadequate resection, in effect?

We hypothesize that these recurrences were also derived from residual peritoneal disease, to reconcile the fact that the recurrences developed in the transplant (1st) graft; however, there is no way to prove this.

In your next patient who presents with ruptured hepatoblastoma, in light of finding peritoneal spread of the cancer 4 years after the first liver transplant, would you consider using ICG to find and remove/destroy peritoneal implants at the time of the liver resection or probably better at the time of reresection or, in view of the likelihood of having tiny metastatic implants too small to see, would you consider doing intraperitoneal chemotherapy at the time of liver resection?

I appreciate your comment regarding the surgical strategy and intraperitoneal chemotherapy. In this specific case, since the patient had received various chemotherapy regimens over several years, we thought the intraperitoneal disease of the HB was resistant to any chemotherapy. In addition, it took 4 years for the lesion to become evident without causing lung metastasis, which led us to hypothesize that the intraperitoneal lesion might have been confined in the right upper quadrant of the abdominal cavity (the lesions visualized by ICG navigation was restricted to only in the right upper abdomen, instead of diffusely spreading peritoneal metastases). Based on these observations, we decided to eliminate intraperitoneal disease surgically, followed by re-transplantation. In general, we treat ruptured HB with intensified intravenous chemotherapy based on risk stratification (in this specific patient, there was no optimal chemotherapy regimen because the initial treatment had been provided 8 years previously). We consider intraperitoneal chemotherapy or surgical removal as the last resort, only after all intravenous chemotherapy regimens are exhausted with no obvious response. We cited a paper regarding this issue and added lines 145-151.

One purported cause of recurrent cancer is that it may be in the blood and be hanging out in the bone marrow. Did you ever check for HB cells in your patient's blood?

Thank you for your input. We did not check for HB cells in the blood. (I am not sure about the reliability of the method for to detecting HB cells in blood). This has been described in lines 168-171.

I would like those questions addressed in your paper.

Fascinating case.

Reviewer 4 Report

The subject matter of this work is laudable and of interest to oncologist and liver surgeon. However, I think that reproducibility is necessary for a scientific article. The authors should think about possibility to give similar treatment in other facilities. But the details of the part of the maneuver of the surgery are unclear. It is necessary to describe the details of surgery techniques to second liver transplantation.

Author Response

The subject matter of this work is laudable and of interest to oncologist and liver surgeon. However, I think that reproducibility is necessary for a scientific article. The authors should think about possibility to give similar treatment in other facilities. But the details of the part of the maneuver of the surgery are unclear. It is necessary to describe the details of surgery techniques to second liver transplantation.

We deeply appreciate your comments.

An additional literature search was performed and several references were added in the discussion. Furthermore, the technical description was modified in lines 95-96, 109-110 and 112-113. 

Round 2

Reviewer 1 Report

The authors have addressed my concerns adequately.

Author Response

The authors have addressed my concerns adequately.

Thank you very much for reviewing our paper and we are glad to satisfy your comment.

Reviewer 2 Report

I would like to thank the authors for their responses

Author Response

I would like to thank the authors for their responses

Thank you very much for reviewing our paper and we are glad to satisfy your comment.

Reviewer 4 Report

 This paper is an important contribution and I recommend that it be accepted for publication.

Ref. no.6 

 This doesn't include the title of the paper.

Author Response

This paper is an important contribution and I recommend that it be accepted for publication.

Ref. no.6

 This doesn't include the title of the paper.

We deeply appreciate your comments and we are very sorry for a mistake. Title of ref.6 have been added.